# COVID-19: The Inflammation Link and the Role of Nutrition in Potential Mitigation

**DOI:** 10.3390/nu12051466

**Published:** 2020-05-19

**Authors:** Ioannis Zabetakis, Ronan Lordan, Catherine Norton, Alexandros Tsoupras

**Affiliations:** 1Department of Biological Sciences, University of Limerick, Limerick V94 T9PX, Ireland; Alexandros.Tsoupras@ul.ie; 2Health Research Institute, University of Limerick, Limerick V94 T9PX, Ireland; Ronan.Lordan@ul.ie (R.L.); Catherine.Norton@ul.ie (C.N.); 3Institute for Translational Medicine and Therapeutics, Perelman School of Medicine, University of Pennsylvania, Philadelphia, PA 19104-5158, USA; 4Department of Physical Education and Sport Sciences, University of Limerick, Limerick V94 T9PX, Ireland

**Keywords:** coronavirus, COVID-19, SARS-CoV-2, inflammation, infection, anti-inflammatory, nutrition, vitamin C, vitamin D, zinc, noncommunicable diseases

## Abstract

The novel coronavirus disease (COVID-19) pandemic caused by severe acute respiratory syndrome coronavirus 2 (SARS-CoV-2) has engulfed the world, affecting more than 180 countries. As a result, there has been considerable economic distress globally and a significant loss of life. Sadly, the vulnerable and immunocompromised in our societies seem to be more susceptible to severe COVID-19 complications. Global public health bodies and governments have ignited strategies and issued advisories on various handwashing and hygiene guidelines, social distancing strategies, and, in the most extreme cases, some countries have adopted “stay in place” or lockdown protocols to prevent COVID-19 spread. Notably, there are several significant risk factors for severe COVID-19 infection. These include the presence of poor nutritional status and pre-existing noncommunicable diseases (NCDs) such as diabetes mellitus, chronic lung diseases, cardiovascular diseases (CVD), obesity, and various other diseases that render the patient immunocompromised. These diseases are characterized by systemic inflammation, which may be a common feature of these NCDs, affecting patient outcomes against COVID-19. In this review, we discuss some of the anti-inflammatory therapies that are currently under investigation intended to dampen the cytokine storm of severe COVID-19 infections. Furthermore, nutritional status and the role of diet and lifestyle is considered, as it is known to affect patient outcomes in other severe infections and may play a role in COVID-19 infection. This review speculates the importance of nutrition as a mitigation strategy to support immune function amid the COVID-19 pandemic, identifying food groups and key nutrients of importance that may affect the outcomes of respiratory infections.

## 1. Introduction—Current Status

Since late 2019, the world has come to terms with the fact that it is facing a major public health crisis against the novel coronavirus disease termed COVID-19 caused by the severe acute respiratory syndrome coronavirus 2 (SARS-CoV-2). As of May 10th, there are over four million confirmed cases of COVID-19 and almost 280,000 deaths globally [1]. The first known case of COVID-19 originated from the city of Wuhan in Hubei Province, China. From there, it has spread to every inhabited continent worldwide. The virus originates from a large class of viruses known as β-coronaviruses common in nature, with many potential natural primary, intermediate, and final hosts [2]. SARS-CoV-2 (Figure 1) is an enveloped positive-sense RNA virus that typically affects the respiratory system, whereby the main known route of transmission occurs due to the spread of droplets generated when an infected person sneezes or coughs or through other mucus environments, including saliva or discharge from the nose [3]. SARS-CoV-2 gains entry to the cell via the angiotensin-converting enzyme 2 (ACE2) receptor [4,5,6,7,8,9,10,11,12], whereby it predominantly infects the lower respiratory tract, binding to ACE2 on alveolar epithelial cells [13]. Upon binding, there is a subsequent response of the immune system via inflammation-related manifestations and recruitment of antigen-presenting cells [10,11,12]. The infectious capacity of the virus in conjunction with the high fatality rates (1%–>5%) has left the world scrambling to bring the pandemic under control [6,7,8,9,14,15]. COVID-19 infection can manifest as an asymptomatic infection, or patients can present with a mild upper respiratory tract illness that may include a cough, chills, fever, fatigue, and shortness of breath [15]. In severe cases, the most common complications are sepsis, acute respiratory distress syndrome (ARDS), heart failure, and septic shock. However, severe viral pneumonia with respiratory failure can potentially lead to death [16]. Multiple organ dysfunction is likely attributable to uncontrolled acute inflammation and cytokine storm release [10,11,12,17,18,19]. As our knowledge of the disease is evolving, it is clear that other symptoms are being identified, including chilblains [20], sudden anosmia or ageusia [21], and even stroke [22].

The inflammatory response plays a crucial role in the clinical manifestations of COVID-19. Post SARS-CoV-2 entry, host factors trigger an immune response against the virus, which, if uncontrolled, may result in pulmonary tissue damage, functional impairment, and reduced lung capacity [10,11]. Damage to the pulmonary interstitial arteriolar walls indicates that an inflammatory response plays an important role throughout the course of the disease, despite the pathogenic effect of the virus [10,11]. Apart from nonspecific inflammatory responses such as edema and inflammatory cell infiltration, severe exfoliation of alveolar epithelial cells, alveolar septal widening, damage to alveolar septa, and alveolar space infiltration has been detected in a distinctly organized manner [10,11]. Thus, SARS-CoV-2 infection can cause pathological changes, degeneration, infiltration, and hyperplasia [10].

Apart from respiratory failure, other common features amongst critical COVID-19 patients include a sudden decline of the patient’s health status approximately two weeks after onset [11], infiltration of monocytes and macrophages into lung lesions, a decrease of lymphocytes such as natural killer (NK) cells in peripheral blood, extremely high levels of inflammatory response due to the proinflammatory cytokine storm, atrophy of the spleen and lymph nodes, along with reduced lymphocytes in lymphoid organs, hypercoagulability, thrombosis, and multiple organ damage [11,24]. These are just a selection of the clinical manifestations that occur as the medical community begins to learn more as the pandemic continues to grow. 

It is therefore apparent that a viral infection-related inflammation and the subsequent cytokine storm in severe cases plays a crucial role in patient outcomes [10,11,12,17]. Furthermore, the coexistence of noncommunicable chronic diseases (NCDs) in COVID-19 patients may aggravate and intensify the inflammatory pathology and increase the risk for adverse outcomes and mortality [25]. This is evident in those at high risk of CVD who may also be at high risk for severe COVID-19 infections due to high angiotensin-converting enzyme 2 (ACE2) expression observed in these patients [4,7,8,9]. Currently, there is a worldwide effort to rapidly diagnose and isolate COVID-19 patients, while scientists congruently search for and repurpose therapies and interventions that are able to counter the most severe effects of the disease [26]. While some therapeutic agents look promising, there is currently no evidence of safety and efficacy in human trials for any proposed treatment for COVID-19 [27]. 

Therefore, an understanding of the mechanisms involved in these manifestations may facilitate the development of targeted anti-inflammatory therapies. However, it is important to recognize that our understanding of the pathomechanisms of COVID-19 are continually evolving. Equally, it is important to determine prophylactic interventions to prevent or lower the incidence of infection. 

To date, the most effective mitigation strategies being implemented globally consist of encouraging standard public health practices such as regular hand washing with soap, wearing a face mask [28], and covering a cough with your elbow [29], along with introducing social distancing measures, “stay in place” guidelines, expansive testing, and contact tracing [30,31]. However, it has been postulated that a healthy nutritional status may support immune function and prevent the onset of a severe infection [32,33,34,35], which leads to the question of whether maintaining a healthy nutritional status is warranted at this time to prime the immune systems of individuals not infected with COVID-19. 

In light of the ongoing public health emergency, a review of pharmacological and nutritional strategies to support the optimization of immune health seems both timely and worthwhile. Therefore, the aims of this review are to give an overview of some of the anti-inflammatory therapies that are under investigation and intended to dampen the inflammatory response in severe COVID-19 infections and to outline dietary considerations implicated in immune function and respiratory illness outcomes, as they may influence patient outcomes in severe COVID-19 infections.

## 2. Mechanisms of the Uncontrolled Inflammatory Response and Cytokine Storm during Disease Progress

The first step in the host cell viral entry is the binding of the viral trimeric spike protein of SARS-CoV-2 to the human receptor angiotensin-converting enzyme 2 (ACE2), which is similar to SARS-CoV-1 infection mechanisms [6,7,8,9]. Although sharing a close evolutionary relationship with SARS-CoV-1, the receptor-binding domain of SARS-CoV-2 differs in several key amino acid residues, allowing for a stronger binding affinity with the human ACE2 receptor, which may account for the greater pathogenicity of SARS-CoV-2 [7,36]. Although ACE2 expression correlates with susceptibility to SARS-CoV-1 infection, the relationship between ACE2 expression levels and the susceptibly to SARS-CoV-2 infection remains unclear [37].

When SARS-CoV-2 infects the respiratory tract, it can cause mild respiratory infections or severe acute respiratory syndrome with consequent inflammatory responses [10,11,12]. During COVID-19 infection, the virus infects antigen-presenting cells, macrophages, and dendritic cells, which present SARS-CoV-2 antigens to T cells, leading to T cell activation, differentiation, and a subsequent enormous cytokine release [10]. The host innate immune system detects viral infections by using pattern recognition receptors (PRRs), including Toll-like receptors (TLR), to recognize pathogen-associated molecular patterns (PAMPs) like lipoproteins, proteins, lipids, and nucleic acids of viral origin [10,38]. 

Subsequently, there is the activation of transcription factors NF-kB, IRF3, and mitogen-activated protein kinases (MAPKs) pathways that induce the expression of inflammatory factors [39]. For example, the binding of SARS-CoV-2 to the TLR causes the release of pro-IL-1β, which is cleaved by caspase-1, followed by inflammasome activation and production of active mature IL-1β, which is a mediator of lung inflammation, fever, and fibrosis [12].

Mast cells (MCs) that are present in the submucosa of the respiratory tract and in the nasal cavity represent a barrier of protection against microorganisms, and they can be virus-activated. Activation of MCs release early inflammatory molecules, such as histamine and proteases, while late activation triggers the production of proinflammatory IL-1 family members, including IL-1, IL-6, and IL-33 [4].

These processes lead to T cell activation and differentiation, including the production of cytokines of various T cell subsets, followed by a massive release of cytokines involved in immune response amplification [10,40]. T cells of the adaptive immune system play an important antiviral role by balancing the battle against pathogens and the risk of overwhelming inflammation or developing autoimmunity [41]. Helper T cells such as the CD4+ promote the production of virus-specific antibodies by activating T-dependent B cells, while other T cells such as the CD8+ are cytotoxic and can kill viral-infected cells [10]. In addition, T helper cells produce proinflammatory cytokines and chemokines via the NF-kB signaling pathway, which, in turn, recruits lymphocytes and leukocytes, such as monocytes and neutrophils, to the site of infection, with a subsequent secretion of large quantities of chemokines and cytokines from all these immune cells to amplify the inflammatory response to the virus infection [10,42]. 

The extensive and uncontrolled release of proinflammatory cytokines is termed the cytokine storm. Clinically, the cytokine storm commonly presents as systemic inflammation and multiple organ failure [11]. Clinical evidence has shown that severe COVID-19 patients have an elevated cytokine profile reminiscent of a cytokine storm as previously observed in SARS-CoV-1 and the Middle East respiratory syndrome (MERS) [11]. The level of inflammatory factors in patients with COVID-19 are increased, among which are IL-1RA, IL-1B, IL-7, IL-8, IL-9, IL-10, granulocyte-macrophage colony stimulating factor (GM-CSF), IFN-γ, fibroblast growth factor (FGF), granulocyte-colony stimulating factor (G-CSF), interferon-γ-inducible protein (IP10), macrophage inflammatory protein 1 alpha (MIP1A), platelet-derived growth factor (PDGF), monocyte chemoattractant protein (MCP1), vascular endothelial growth factor (VEGF), and tumor necrosis factor (TNF-α). Furthermore, levels of IL-2, IL-6, IL-7, IL-10, IP10, G-CSF, MCP1, MIP1A, and TNF-α were higher in critically ill patients with COVID-19 than that in mild groups [11,12,43].

In the majority of severe COVID-19 patients, the cytokine storm was associated with high levels of erythematosus sedimentation rate (ESR) and C-reactive protein (CRP), which were associated with ARDS, hypercoagulation, and disseminated intravascular coagulation (DIC), presenting as thrombocytopenia, thrombosis, and gangrene of the limbs. It is possible that the cytokine storm may exacerbate lung damage [11,44]. Blanco-Melo et al. [45] determined that the unique inappropriate inflammatory response of COVID-19 is due to the low levels of type I and II interferons in conjunction with an elevated expression of IL-6 and increased chemokines. They hypothesize that the enhanced inflammatory response, along with a reduced innate antiviral defense, may be the “defining and driving feature” of COVID-19 infections. 

Although the lungs are considered the main target organ of SARS-CoV-2, the virus can affect many other organs, including the heart and blood vessels, the kidneys, the gut, and brain, through various mechanisms [46]. It has been widely proposed that one of the ways the virus can critically affect these organs is through an intense inflammatory reaction, while other mechanisms have yet to be elucidated. For instance, it is now documented that COVID-19 patients may be predisposed to arterial and venous thromboembolisms due to excessive inflammation, diffuse intravascular coagulation, and hypoxia. In a Dutch study of COVID-19 patients, one-third exhibited blood clots [47]. Therefore, antiplatelet agents may be considered to treat COVID-19. Overall, considering inflammation plays a significant role in COVID-19 pathology, it would seem logical that it would be important to control cytokine production, given that they are responsible for the accumulation of immune cells and fluids. As such, anti-inflammatory treatments may hold promise for the management of COVID-19 complications.

## 3. Current Knowledge of COVID-19 Treatment and Anti-Inflammatory Approaches

Classic first-line antiviral treatments are a potential therapeutic against COVID-19, which, when concomitant with organ function support, are very important to reduce mortality for mild and critical patients [11,48]. Antiviral therapy against SARS-CoV-2 consists of using different polymerase inhibitor drugs that are currently on the market and approved for use against other viruses; these include Ribavirin, Remdesivir, Sofosbuvir, Galidesivir, and Tenofovir [48]. These antiviral drugs have previously exhibited anti-inflammatory properties, either individually or combined as highly active antiviral therapies. However, the long-term use of some of these antiviral therapeutics against other persistent viral infections has been associated with inflammation-related cardiovascular side effects [49,50]. 

Researchers are also targeting inflammation through the investigation of immunomodulatory and anti-inflammatory therapies to reduce systemic inflammation before the onset of multiorgan dysfunction [11,51]. Therefore, biological agents targeting cytokines expression and specific cytokine antagonists, such as IL-6R monoclonal antibodies, TNF inhibitors, IL-1 antagonists, Janus kinase inhibitor (JAK) inhibitors, etc., have been considered [11]. Adopting an approach against specific cytokines entails the danger of only inhibiting one aspect of the inflammatory pathways involved. As a consequence, it may not be very effective in curbing the cytokine storm in COVID-19, as various cytokine pathways are of significant importance, and immunosuppression may actually compromise host defenses [52,53]. Furthermore, some of these specified anti-inflammatory medications, such as JAK inhibitors, may also block the production of the antiviral interferons such as the INF-a, which may have negative consequences for the immune response [11,54].

Researchers are also investigating the possible effects of antimalarial drugs such as chloroquine and hydroxychloroquine against SARS-CoV-2. These molecules are generally prescribed for autoimmune diseases such as systemic lupus erythematosus and rheumatoid arthritis in patients whose disease status has not improved with other treatments. However, chloroquine and hydroxychloroquine possess a broad spectrum of antiviral effects against several viral infections, including coronaviruses such as SARS-CoV-1 [11,37,48,49,55,56]. In vitro experiments in China identified chloroquine as a promising therapeutic against SARS-CoV-2 [56,57], while the immunomodulatory effects of its derivative hydroxychloroquine may be more effective at targeting the cytokine storm that occurs in the late phase of critically ill COVID-19 infected patients, with less side effects [55,58].

Hydroxychloroquine interferes with lysosomal activity and autophagy and alters transcription and signaling pathways, which can result in the modulation of cytokine production [59], all of which are postulated to dampen the effects of the proinflammatory cytokine storm of severe COVID-19 patients. Furthermore, hydroxychloroquine has been reported to have better outcomes when combined with other drugs, such as antibiotics like azithromycin, or drugs used for rheumatoid arthritis, such as tocilizumab/atlizumab, in addition to the standard medical management for septic shock and ARDS [60]. Notably, some antibiotics, including azithromycin, have exhibited anti-inflammatory potency [51,61,62], while tocilizumab/atlizumab contains a humanized monoclonal antibody against the IL-6 receptor; thus, it is mainly used for reducing inflammation during autoimmune disorders [63].

Hydroxychloroquine, in combination with azithromycin, has been particularly focused upon due to the results of a French study where 26 COVID-19 patients received the combination treatment versus control groups. However, there were several issues with the study design, including its small sample size, the fact that the control groups were from different hospitals, the study was not blinded, and a myriad of other issues [64]. While the study provided an indication that hydroxychloroquine was worth further investigation, its results have been blown out of proportion in the media. Hydroxychloroquine has even been prematurely touted as a “game-changer” by President Donald Trump of the United States, who has admonished that he may even consider taking this untested drug against COVID-19 [65]. Some countries have allowed compassionate use of these drugs [66,67]. The promotion of hydroxychloroquine or chloroquine without substantial evidence of randomized, controlled trials is a significant safety and efficacy concern and requires further intensive investigation [68,69,70]. Currently, a study from the United States Veterans Health Administration indicated that patients administered hydroxychloroquine alone or in combination with azithromycin were no less likely to require mechanical ventilation and had higher mortality when on hydroxychloroquine alone versus the standard treatment [71]. This example demonstrates the necessity for world leaders to consult with their experts prior to promoting unproven medications against SAR-CoV-2 as a matter of public safety until further research is conducted.

Several researchers mention the relationship of anti-CVD-related drugs with COVID-19. Indeed, there is a focus on the anti-inflammatory drug colchicine, which has been previously used effectively against cardiovascular disorders [72]. Since ACE2 is implicated in COVID-19 infection, the role of angiotensin-converting enzyme inhibitors (ACEIs)/angiotensin receptor blockers (ARBs) of the ACE2 receptor and its rennin-angiotensin system, which are typically used for hypertension, has recently been evaluated [7,8,9,37]. ACEIs/ARBs perform a protective role in the cardiovascular system by also increasing the expression ACE2 in the heart [37]. However, the impact of ACEIs/ARBs on ACE2 in other organs, especially whether they could influence the expression level and activity of ACE2 in the lungs, with a subsequently higher susceptibility to SARS-CoV-2 infection, remains unknown. Thus, if ACEIs/ARBs have the capacity to upregulate the expression and activity of ACE2 in the lungs, they may play a dual role in COVID-19. On one hand, the higher level of ACE2 might increase the susceptibility of cells to SARS-CoV-2 viral host entry and propagation, whereas, on the other hand, the activation of ACE2 might ameliorate the acute lung injury induced by SARS-CoV-2 [37,73,74]. Despite these concerns, the European Society of Cardiology recommends that patients continue their usual antihypertensive medications due to lack of evidence [75]. This also may have dietary implications due to the modulatory effects dietary patterns can have on hypertension but also due to the fact that some foods are associated with high levels of ACE inhibitory peptides [76]. Thus, further research is required to investigate the role of ACE2 activation, expression, and its related pathways in COVID-19.

As aforementioned, implementing an anti-inflammatory strategy is challenging, as it is not yet clear if any specific features of the immune response can be inhibited directly without compromising a patient’s overall immune defense [52]. It is important to determine the optimum method to reduce inflammation. Further research is also required to gain an understanding of the temporal features of the COVID-19 inflammatory response and to determine at what stage of the infection should pharmaceutical treatments be administered. Similarly, an understanding of the dosing and sexual dimorphism in drug metabolism and disease presentation is required. Likewise, ethnicity may be a determining factor in severe COVID-19 infections that requires further investigation [77,78].

Furthermore, considering an increased risk of clot development can turn mild cases into more severe and life-threatening emergencies, a preventative approach targeting thrombosis should also be considered [47], particularly those that may also target inflammation [79]. Reactive oxygen species (ROS) also play a crucial role in the inflammatory response. As such, utilizing compounds with antioxidant properties may also be considered to reduce the cytokine storm induced by the viral infection [80]. Indeed, antioxidative therapies are being considered for ameliorating cardiac injuries in critically ill COVID-19 patients [81].

Among other measures, the nutritional status of an infected patient should also be considered, as nutritional deficiencies may increase a patient’s risk to developing a severe infection of COVID-19 [82]. Introducing nutritional interventions should not be overlooked due to their potential for beneficial clinical outcomes [83]; in particular, consideration must be given to intensive care unit (ICU) patients [84]. 

## 4. Noncommunicable Diseases and COVID-19—The Link between Immunity and Nutrition

On 23rd March 2020, the World Health Organization (WHO) issued guidelines to highlight that those diagnosed with noncommunicable diseases (NCDs) may be more susceptible to developing COVID-19 (Figure 2) [25,85].

The link between viral infection severity and NCDs has been observed in other viral infections, such as influenza [86]. Chronic and unresolved inflammation is implicated in the onset, progression, and development of NCDs [87,88]. Furthermore, it is thought that underlying systemic inflammation may exacerbate COVID-19 infection [85]. For instance, in COVID-19 patients with underlying CVD, the respiratory symptoms of infection seem to be more severe [89]. Data from the United States Centers for Disease Control and Prevention (CDC) demonstrates the high risk of mortality in individuals with pre-existing health conditions [90]. Considering patients with pre-existing conditions such as diabetes and high blood pressure exhibit damage to the blood vessels, the existence of increased thrombotic complications in COVID-19 may account for the higher risk of serious disease in these patients [46]. Certainly, diabetes is a significant risk factor for COVID-19 [91] and is associated with a two-fold increase in mortality and potential disease severity [92].

Generally, where possible, an effective strategy to reduce one’s risk of developing NCDs is to control the activities of inflammatory mediators via modifiable risk factors such as diet, exercise, and healthy lifestyle choices [87,93]. Only the adoption of a long-term and consistent dietary pattern benefits human health. Conversely, adoption of an unhealthy diet and lifestyles is associated with low-grade inflammation and increased oxidative stress, which could lead to the development of NCDs [87,94]. Certainly, there is considerable evidence that the food and nutrients we consume affect how our immune system functions [34,95,96]. The role of nutrition amidst the current global pandemic will be discussed in Section 5.

Some reports [25,90] have suggested that a higher body mass index (BMI) or excess adiposity carry risk factors for complications arising from COVID-19 infection. This may be due to a higher prevalence of pulmonary problems in obese populations relative to their healthy weight counterparts [97]. Patients with obesity and comorbidities that compromise their heart or lung function are likely at higher risk for developing severe diseases with COVID-19 [14], much like the nonobese patients with those risk factors. Maintaining a body weight and composition in line with recommendations [98] for stature and gender (among other considerations, such as comorbid disease or athletic discipline) is prudent.

## 5. Immunomodulatory and Anti-Inflammatory Potential of Maintaining a Healthy Nutritional Status

Nutritional status can have a significant impact on an individual’s overall health, the reduction of NCDs, and a reduced susceptibility to developing infections. In this section, the beneficial effects of a healthy diet are discussed in relation to the current COVID-19 pandemic. However, it must be noted that, to date, there are no known evidence-based therapeutics or treatment strategies available to prevent the incidence or severity of COVID-19 infection. Likewise, there is no single food or natural remedy that has been proven to prevent COVID-19 infections, which has been made clear by the WHO [99]. However, learning from previous research in relation to other viral infections, it is clear that nutritional status plays a significant role in patient outcomes [100]. The presence of comorbidities in COVID-19 patients is currently a significant concern [25], which leads to the question of whether the nutritional status of these patients is a concern. Likewise, for those not affected, could following a diet characterized by anti-inflammatory properties potentially benefit or prevent severe infections in patients with comorbidities who contract COVID-19? In general, how important is it to follow a healthy diet currently for the general population? There are various diets and nutrients that potentially impart anti-inflammatory and immunomodulatory properties on diseases, including cardiovascular diseases, lung diseases, and numerous NCDs, without risking immunosuppression [101,102,103,104]; thus, diet is worthy of significantly more research. 

Due to its safety and ease of application, nutrition is well-placed to have a key role in the challenge of “keeping healthy people healthy”. Moderate-quality evidence suggests that dietary patterns and individual nutrients can influence systemic markers of immune functions. However, the complexity of the interaction (nutrition and immunology) necessitates further research in advance of population–based dietary recommendations. At a minimum, the attainment of reference nutrient intakes (RNIs) or recommend daily allowance (RDA) for those nutrients thought to have a role in supporting immune functions is recommended at this time. These nutrients and their potential roles in either immune functions or respiratory tract infections are discussed below. 

Maintaining nutritional status at this time is significant, since the battle against COVID-19 will likely last longer than what was initially anticipated. It is also apparent that social isolation and mitigation measures such as “stay at home” orders will last for a prolonged period for millions of people around the world. Therefore, in order to prevent the development or exacerbation of NCDs, which are currently a significant burden on global health systems [105], and to maintain a healthy immune system, special attention must be given to maintaining a healthy diet, lifestyle, exercise regime, and minimal stress as much as safely possible at this difficult time [106].

Indeed, it is particularly important at this time to consider our elderly communities, as the elderly are not only predisposed to NCDs, but they are also vulnerable to the increased risk of malnourishment, infections, and COVID-19 [90]. In fact, age itself is a risk factor for developing COVID-19 [107]. This is due to a functional decline of the immune system with age known as immunosenescence [108,109]. Malnourishment can occur for several reasons, including poor socioeconomic conditions, mental status, social status, and a host of other multifactorial issues [110]. Indeed, often, there are nutritional deficiencies of calcium, vitamin C, vitamin D, folate, and zinc amongst elderly populations [111]. Malnourishment can exasperate an impaired immune system in the elderly, making them susceptible to infections [112]. A healthy, balanced diet can offer the necessary macro- and micronutrients, prebiotics, probiotics, and symbiotics in the elderly that can restore and maintain immune cell function, thus increasing protection against chronic inflammation-related NCDs, on the one hand, and potential infections and related inflammatory manifestations on the other hand [87,113]. The foods, food groups, and nutrients described in this review have been chosen due to their potential anti-inflammatory and immunomodulatory properties. However, there are other nutrients, etc., worthy of further investigation, including vitamin A, selenium, and various probiotics and nutraceuticals [114,115].

### 5.1. Dietary Patterns

#### 5.1.1. Mediterranean Diet

Various dietary patterns have been linked to the risk of inflammatory conditions [116] and respiratory disease [117]. One diet synonymous with anti-inflammatory properties is the Mediterranean diet, which is characterized by a relatively high dietary intake of minimally processed fruit, vegetables, legumes, olive oil, whole grains, nuts, and monounsaturated fats, followed by low-to-moderate consumptions of fermented dairy products, fish, poultry, wine, and, lastly, low consumptions of processed and red meats [87,118]. A balanced diet rich in these foods is associated with anti-inflammatory and immunomodulatory compounds, including essential vitamins (C, D, and E) and minerals (zinc, copper, calcium, etc.), that affect a person’s nutritional status [119]. 

Several foods associated with the Mediterranean diet and other healthy dietary patterns contain bioactive compounds that go beyond just vitamins and minerals, including bioactive phenolic compounds; polar lipids; and peptides with potent anti-inflammatory, antithrombotic, and antioxidant properties. These molecules can synergistically act to prevent and protect against inflammatory manifestations and associated thrombotic and ROS-related complications [87,120,121,122,123,124].

Therefore, healthy dietary patterns such as the Mediterranean diet or similar are beneficial against NCDs but, potentially, also against infections such as COVID-19 due to their effects on immune health [87,125,126]. The traditional Mediterranean diet has been found to have protective effects for allergic respiratory diseases in epidemiological studies [127] and for inflammatory conditions [116,128]. 

Overall, much of the benefits of the Mediterranean diet has to do with the high intake of a wide range of bioactive compounds that come from all aspects of the diet [87,129,130] and the relatively low intake of overly processed foods associated with negative health effects, such as that of the Western diet [131]. Of course, it is the ideals of the Mediterranean diet that are important to follow, as a traditional Mediterranean diet is practically nonexistent these days, and countries of the Mediterranean follow a diet more closely related to the Western diet than that of their ancestors [129]. Most national dietary guidelines follow similar principals to the Mediterranean diet by promoting the intake of fruit and vegetables, etc., in large quantities and advise that people limit their intake of processed foods. 

#### 5.1.2. Western Diet

The “Western” dietary pattern, prevalent in developed countries, is characterized by high consumptions of processed foods such as refined grains, cured and red meats, desserts and sweets, deep-fried foods, and high-fat products [132]. As a result, there is a high intake of sugars, saturated fats, and refined carbohydrates that may be associated with hyperglycemia and hyperlipidemia. Consequently, there is a high prevalence of type II diabetes mellitus and obesity, thus potentially putting these particular populations at increased risk for severe COVID-19 complications [133]. The Western diet is highly associated with hyperglycemia and advanced glycation end products (AGEs) that are linked with inflammation, metabolic complications, and chronic disease [134]. Notably, hyperglycemia is a risk factor associated with high mortality in patients with severe COVID-19 infection [135]. This pattern of intake has also been associated with an increased risk of asthma in children [136]. In adults, a Western diet has been shown to be positively associated with an increased frequency of asthma exacerbation [137]. This dietary pattern is also associated with other inflammatory conditions [138] due to the relative absence of the beneficial nutrient components that are present in the Mediterranean diet and other healthy dietary patterns. Similarly, before the pandemic, the advice was that people should avoid following dietary patterns consistent with the Western diet due to the profound negative health effects associated with its adherence, including the development of NCDs and the promotion of a low-grade inflammatory state. 

### 5.2. Foods, Food Groups, and Individual Nutrients

#### 5.2.1. Fruit and Vegetables

Fruit and vegetable intakes have been investigated for potential benefits in association with respiratory [139] and inflammatory [140] conditions due to their nutrient profile consisting of antioxidants, vitamins, minerals, and phytochemicals that include phenolic compounds that can exert antioxidant, anti-inflammatory, and other beneficial effects [123,141,142]. Indeed, polyphenols may also exhibit antiviral effects against the West Nile virus, Zika virus, and Dengue virus [143,144]. A recent umbrella review of research on fruits, vegetables, and health [145] provides support for the dietary recommendation “to consume four or more servings per day” in this context; however, excessive intakes may also displace other valuable foods, potentially leading to nutrient deficiencies of some vitamins or minerals not found in these foods. For example, zinc is more prevalent in meat or dairy products. Fruits and vegetables are also sources of fiber important for gut health. Many of the beneficial components of fruits and vegetable will be further discussed in Section 5.3. 

#### 5.2.2. Fish and Fish Oils

Fish and fish oils have been associated with various health benefits against numerous NCDs, including CVD and cancer [146,147,148,149]. Omega-3 polyunsaturated fatty acids (PUFA), namely eicosapentaenoic acid (EPA) and docosahexaenoic acid (DHA) from fish, other marine sources, and supplements, have been shown to be anti-inflammatory through several cellular mechanisms, including their incorporation into cellular membranes and resulting altered synthesis of eicosanoids [146,147]. Li et al. [148] reviewed 89 systematic reviews and meta-analyses investigating fish intake and all-cause mortality, concluding that fish intake at 2–4 servings per week is associated with the largest risk reduction. 

However, despite these potential positive effects, there is conflicting evidence in relation to fish or fish oil consumption for some viral infections. In influenza models, fish oil-fed mice demonstrated impaired resistance to influenza infection due to their immunomodulatory and anti-inflammatory properties, which negatively dampened the immune response of the mice against the infection [150]. Indeed, in another mouse model, fish oil intake delayed influenza virus clearance and impaired the immune response in the lungs of mice by disrupting interferon-γ and immunoglobulin A [151]. A further study indicated that it may be due to the impairment of virus-specific T lymphocyte cytotoxicity [152]. However, substantial human trials in this area are lacking. In contrast, EPA and DHA may act as substrates for the synthesis of specialized pro-resolving lipid mediators such as maresins, resolvins, and protectins, of which protectins may reduce the replication of influenza [153] and potentially affect the inflammatory manifestations of respiratory viral diseases [154]. Although their synthesis via fish oil supplementation in humans requires further investigation, as there are differing reports of their clinical relevance [79,155,156].

Prostaglandins, thromboxanes, and leukotrienes are proinflammatory molecules formed from arachidonic acid (AA). Suppressing prostaglandin synthesis by targeting the cyclooxygenase enzymes (COX-1 and COX-2) with nonsteroidal anti-inflammatory drugs (NSAIDs) reduces inflammation, pain, and fever. However, using COX-1/2 inhibitors in some viral infections such as influenza A may have differential and potentially negative effects, as demonstrated in vivo [157]. Previously, it was shown that SARS-CoV-1 could bind with the COX-2 promotor, increasing its expression [158]. However, it is not currently known whether NSAIDS may be beneficial or lead to more severe COVID-19 symptoms, as prostaglandins such as prostacyclin (PGI_2_), prostaglandin E_2_ (PGE_2_), and prostaglandin D_2_ can both reduce and induce inflammation [159]. Certainly, it has been suggested that selective inhibition human microsomal prostaglandin E synthase 1 (mPGES-1) may be a viable therapeutic target for SARS-CoV-2 [160]. While it is known that n-3 fatty acids can beneficially interact with the COX enzymes [146,161], it also is not clear if fish or fish oil consumption may be beneficial against SARS-CoV-2 infection.

In terms of dietary strategies, it is known that marine n-3 PUFA can replace AA in the phospholipids of cell membranes, thus affecting eicosanoid and prostaglandin synthesis. Indeed, the supplementation of EPA and DHA raises the level of these fatty acids in the phospholipids of cells involved in inflammation in a time and dose-dependent fashion at the expense of AA [146]. There is the potential that AA, EPA, DHA, and other dietary unsaturated fatty acids can inactivate enveloped viruses. It is thought that these fatty acids and others cause leakages or lysis of the viral envelopes by disrupting the membrane integrity, amongst other potential mechanisms [162,163,164,165]. Indeed, alveolar immune cells such as macrophages, leukocytes, natural killer cells, and B and T cells release unsaturated fatty acids such as AA into the surrounding microenvironment when challenged by viruses, including SARS-CoV-1, MERS, and, potentially, SARS-CoV-2 [162]. Notably, an in vitro model of human cells (Huh-7 and VeroE6) infected with a human coronavirus (HCoV-229E) demonstrated that several bioactive lipids downstream of phospholipase A_2_ (PLA_2_) activation were upregulated by the host cells. It is postulated that coronaviruses modulate the host lipid profile to optimize and maintain a specific homeostasis for viral replication. However, exogenous supplementation of AA and linoleic acid suppressed viral replication by interfering with the optimal host lipid conditions for viral replication. Notably, exogenous supplementation of AA and linoleic acid was also conserved when human cells were infected with MERS [166]. EPA, DHA, and AA also inhibited the replication of enterovirus A71 and coxsackievirus A16 [167]. While it is suggested that the oral or intravenous administration of various bioactive lipids could potentially reduce the severity and/or enhance the recovery of those infected with COVID-19 [162], a dietary prophylactic approach or a dietary strategy for recovering patients is also worth considering. Further research is certainly required, as increasing AA via the diet might seem counterintuitive, as it is mainly proinflammatory, and it is associated with the Western diet [168], but EPA and DHA could be considered.

Most fish oil studies were conducted using EPA and DHA, which have previously exhibited immunomodulatory effects [169], but it is worth noting that other lipid molecules in fish, including polar lipids, exhibit anti-inflammatory effects through differing mechanisms by modulating the activities and metabolism of the potent proinflammatory and prothrombotic mediator platelet-activating factor (PAF) [146,170]. Certainly, PAF and its receptor (PAF-R) are known to be involved several NCDs [87] and viral infections such as HIV [50], dengue virus [171,172], respiratory syncytial virus [173], and lung injury caused by influenza A [174]. Indeed, a lipidomic analysis of in vitro human cells infected with a coronavirus (HCoV-229E) demonstrates a 3.5-fold elevation of PAF levels [166]. Blockade of the PAF-R pathways with PAF antagonists prevented severe disease manifestations in the dengue virus [171]. Similarly, PAF-R antagonism in mice suffering influenza A induced protection against lung injury and mortality [174]. In addition, evidence in humans shows that highly active antiretroviral therapy with anti-PAF effects may attenuate PAF metabolism in the blood of HIV patients [49,50]. PAF-R antagonists have also been investigated for their anti-inflammatory effects in HIV-1-associated neurocognitive disorders [175]. Therefore, dietary PAF inhibitors such as polar lipids in fish [146], or even other dietary sources [147,176,177,178], may exhibit anti-inflammatory effects that may be beneficial in viral diseases and NCDs, as opposed to PUFA esters or fatty acids that act through differing mechanisms.

Fish and other foods also contain polar lipids that exhibit potent antithrombotic effects against PAF and other prothrombotic pathways, including thrombin, collagen, and adenosine diphosphate (ADP) [146,179,180]. Other bioactive compounds in fish, such as peptides, may also prevent against thrombosis, the generation of ROS, and hypertension [124,181]. Therefore, increasing fish consumption may provide the nutrients and bioactive molecules that may influence some of the pathomechanisms and complications of COVID-19, such as inflammation and thrombosis. Currently, 2–4 g of n-3 fatty acids seem to be physiologically relevant against hypertension, inflammation, and thrombosis; however, even higher doses have been suggested [146,182,183,184]. Studies are warranted to discern whether fish consumption may beneficially support immune function against COVID-19. 

#### 5.2.3. Vitamin C

Vitamin C (ascorbic acid) is a water-soluble vitamin, the consumption of which has been part of cultural practice when suffering with a cold or flu for decades. This is due to research published by Nobel prize winner Linus Pauling (circa 1970) theorizing how vitamin C helps to treat colds [185]. An analysis of 29 studies including 11,306 participants concluded that supplementing with 200 mg or more of vitamin C does not reduce the risk of contracting a cold [186,187]. However, regular vitamin C supplements had several benefits, including:Reduced cold severity: They reduced the symptoms of a cold, making it less severe.Reduced cold duration: Supplements decreased the recovery time by 8% in adults and 14% in children, on average [187].

Indeed, a supplemental dose of 1–2 g was enough to shorten the duration of a cold by 18% in children, on average [186]. Despite much of the research on vitamin C intake being conducted using supplements, vitamin C can also be obtained through dietary sources such as citrus fruits, berries, brassicas, leafy greens, tomatoes, and various other fruits and vegetables. Fresh fruit consumption high in vitamin C, even at low levels, was associated with a reduction of wheezing symptoms in children in a cross-sectional study [188]. Interestingly, most studies examining vitamin C intakes fail to account for dietary vitamin C intakes in their trial design [189], potentially contributing to the disparity of outcomes observed at a grand scale [186]. As reviewed by Hemilä [190], there is a dose-concentration relationship, whereby healthy individuals may not benefit from supplementation if the dietary intake of vitamin C is 200 mg/day. However, this does not apply to all situations and certainly not in the case of patients with infections, as they have lower vitamin C levels due to alterations in the metabolism [190]. Although evidence is currently weak in relation to the utility of vitamin C against COVID-19 infections [191], there was sufficient evidence from investigations of vitamin C in other respiratory diseases to begin conducting clinical trials for severe hospitalized cases (www.clincialtrials.gov: NCT04264533). Currently, the recommended daily allowance (RDA) of vitamin C for healthy adults according to the United States National Institute of Health (NIH) dietary reference index (DRI) is 75–90 mg/d (tolerable upper intake level is 2 g/d) [192]. Due to a lack of evidence against COVID-19, there are limited recommendations for vitamin C intake. However, previously, doses of 1–2 g/d were effective in preventing upper respiratory infections. As those levels are not attainable through dietary sources, supplementation may be advised for those at a higher risk of respiratory infections. However, as previously discussed, doses above 200 mg/day might not benefit healthy individuals.

#### 5.2.4. Vitamin D

The data on vitamin D and immune function is also equivocal. Vitamin D is often referred to as the sunshine vitamin; it is also found in eggs, mushrooms, fatty fish such as salmon, milk and dairy products, or foods fortified with vitamin D. A recent analysis of 25 randomized controlled trials of 11,000 patients showed an overall protective effect of vitamin D supplementation against acute respiratory tract infections [193]. The research is not conclusive, and some studies of vitamin D have not shown any benefit. On balance, Vitamin D supplementation was safe, and it protected against acute respiratory tract infection overall. Patients who were very vitamin D-deficient and those not receiving bolus doses experienced the most benefits. Recent research has suggested that increasing vitamin D intakes may reduce the risk of infections and, also, COVID-19 [33]. The authors speculate that low vitamin D levels may play a role in the high incidences of COVID-19, as the outbreak occurred in winter. Indeed, several research groups have speculated the same and considered whether latitude plays a role in infection susceptibility [194]. Researchers have also highlighted the need for vulnerable groups to maintain their vitamin D status to reduce the risk of respiratory infections, including COVID-19, through various mechanisms [195,196]. Moreover, a trial is currently underway (www.clinicaltrials.gov: NCT04334005) with the aim of assessing the utility of vitamin D as an immune-modulating agent. Patients will be monitored to see if there is an improvement of health status in nonsevere symptomatic patients infected with COVID-19, as well as determining if vitamin D could prevent patient deterioration. Vitamin D itself may exhibit antiviral effects by interfering with viral replication and through its immunomodulatory and anti-inflammatory properties [197]. As a result, it is assumed that increased vitamin D may provide benefits against SARS-CoV-2 infection [198]. Patients in Switzerland that tested positive for SARS-CoV-2 had significantly lower 25-hydroxyvitamin D levels (*p* = 0.004) (median value 11.1 ng/mL) compared with negative patients (24.6 ng/mL), which was also confirmed by stratifying patients according to age >70 years [199]. However, a recent analysis of the UK Biobank provided no evidence to support a potential role for the concentration of 25-hydroxyvitamin D as an explanation for the increased susceptibility of individuals positive for COVID-19 infection. However, they did discern that COVID-19 disproportionately affected Blacks and minority ethnic populations independent of vitamin D levels [78].

Consequently, further research is required before any determination can be made about the prophylactic or therapeutic values of vitamin D against COVID-19. The RDA of vitamin D for healthy adults according to the NIH DRI is 15–20 µg/d (600–800 IU; tolerable upper intake level 100 µg/d or 4000 IU) [200]. However, researchers recommend doses of 10,000 IU/d (250 µg/d) of vitamin D_3_ for several weeks to quickly raise 25-hydroxyvitamin D concentrations, followed by 5000 IU/d (125 µg/d) in order to maintain concentrations above 40–60 ng/mL (100–150 nmol/L), which may be beneficial against COVID-19. Higher doses of vitamin D are certainly advisable for vulnerable individuals. It is also particularly important to increase dietary sources of vitamin D at this time, as many globally are subjected to less sun exposure due to “stay at home” mitigation strategies. Although it is unclear whether the extreme doses suggested would be beneficial against COVID-19, it is clear that they are unattainable through dietary means and would require supplementation.

#### 5.2.5. Vitamin E

Vitamin E is a group of fat-soluble antioxidants that includes molecules such as tocopherols and tocotrienols. However, α-tocopherol is the only form of vitamin E recognized to meet human requirements. α- and γ-tocopherols are both abundant forms of dietary vitamin E. However, α-tocopherol is approximately 5 to 10-fold higher than γ-tocopherol in circulation due to the differences in bioavailability and metabolism [32]. Various foods provide vitamin E: nuts, seeds, and vegetable oils are significant contributors to dietary intakes, as well as green leafy vegetables and fortified cereals [32,201]. Numerous animal and human studies reviewed by Wu and Meydani [202] have indicated that vitamin E deficiency impairs both humoral and cell-mediated immune functions. It is generally accepted that vitamin E may exert its immunoenhancing effects by scavenging oxygen species to reduce oxidative stress [201], and it may induce anti-inflammatory effects [202]. Vitamin E can also protect the polyunsaturated fatty acids (PUFAs) in the cell membranes from oxidation, regulate the production of ROS and reactive nitrogen species (RNS), and modulate signal transduction. Likewise, vitamin E is present in high concentrations in immune cells, which protects them from oxidative damage due to their high metabolic activity and PUFA contents [32,203]. Notably, aging is associated with dysregulation of the immune system, predisposing people to increased oxidative stress and inflammation. This leads to increased incidences of infections in the elderly, such as influenza. Despite the fact that the elderly have comparable vitamin E levels to younger individuals, increasing vitamin E intakes may benefit their immune function, impart resistance to infection, and reduce morbidity due to infections [204,205,206]. As the elderly are predisposed to infection due to immunosenescence [207], it is worth investigating vitamin E for potential benefits against COVID-19. Indeed, it has been suggested that a combination of vitamins C and E may be a useful antioxidant therapy for cardia complications of COVID-19 [81]. However, there is little evidence to date on the utility of vitamin E as a prophylactic or therapeutic agent against COVID-19. The RDA of vitamin E for healthy adults according to the NIH DRI is 15 mg/d (tolerable upper intake level 1000 mg/d) [192]. While vitamin E has been recommended as a potentially beneficial nutrient against COVID-19 infection [82], there are currently no estimates of a beneficial dosage. 

#### 5.2.6. Zinc

Zinc is a trace element of the diet that is critical to the development of immune cells and an important cofactor for many enzymes [208]. Zinc deficiency can contribute to defective cell-mediated immunity and to increased susceptibility to various infections, including pneumonia [209,210]. Despite various foods containing zinc, including meat, dairy, and legumes, etc. [211,212,213], the majority of dietary zinc research has focused on zinc supplementation. Therefore, it is not clear from dietary studies high in zinc if increased zinc intake can protect against the effects of viral infections. Indeed, zinc supplements and lozenges are a popular remedy for fighting off colds and respiratory illnesses. Some studies have found that zinc lozenges may reduce the duration of a cold by about a day and may reduce the number of upper respiratory infections in children [214]. The data on zinc are mixed, and further research is necessary to support a supplementation policy. It has certainly been suggested that increasing zinc intakes may be useful against COVID-19 infections by reducing viral replication and reducing the effects of the gastrointestinal and lower respiratory symptoms [89]. While the RDA of zinc according to the NIH DRI is 8–11 mg/d of zinc for adults (tolerable upper intake level 40 mg/d) [215], it has been suggested that a zinc intake of 30–50 mg/d might aid in the control of RNA viruses such as influenza and coronaviruses [115]. 

#### 5.2.7. Copper

Copper is recognized as an essential trace element; food groups such as offal and nuts and, to a lesser extent, cereals and fruits can be regarded as suitable dietary sources [216]. Copper is one of many dietary minerals, including zinc, that are essential to maintaining the integrity of DNA by preventing oxidative DNA damage [217]. Copper deficiency has been associated with altered immune responses and an increased frequency of infections [218]. Mechanistically, copper deficiency can occur following chronic TNF-α-induced inflammation of the lungs, and it is proposed that copper supplementation may ameliorate this lung inflammation in mice [219]. However, high levels of serum copper have been associated with adverse effects on respiratory health in humans [220]. Regardless of these inconsistencies, Bost et al. and ViXas et al. [216,221] have presented reviews relating to the adequacy of copper intake in developed countries. They have found that adults and elderly populations generally meet dietary recommendations. As a result, a widespread adherence to healthy eating guidelines should achieve sufficiency. The RDA of copper according to the NIH DRI is 900 µg/d for adults (tolerable upper intake level 10 mg/d) [215]. While there has been no recommended dietary intake of copper against COVID-19, a copper intake of 7.8 mg/d has been shown to reduce oxidative stress and alter immune function, albeit it is unknown whether those changes were beneficial [222]. Therefore, considerably more research is required to determine whether higher intakes of dietary copper may benefit immune functions against viral infections. 

#### 5.2.8. Fiber

Studies demonstrate a lower incidence of bacterial translocation across the gut barrier with the administration of dietary fiber [223], suggesting that this nutrient modulates immunity. Among the potential mechanisms by which dietary fiber influences the immune system are changes to the gut-associated lymphoid tissues (GALT) arising from altered gut microflora. Prebiotic fiber is neither hydrolyzed nor absorbed in the upper part of the gastrointestinal tract and becomes a selective substrate for one or a limited number of beneficial colonic bacteria [224,225]. Indeed, fiber intake has been shown to increase the survival of influenza-infected mice via various mechanisms, including blunting the immune response by altering the type of immune cells generated and the generation of diet-derived short-chain fatty acid (SCFA)-enhanced CD8^+^ T cell effector functions by altering the cell’s metabolism [226]. Western diets are often deficient of dietary fiber [138]; therefore, increasing the consumption of both soluble (oat bran, barley, nuts, seeds, beans, lentils, peas, and some fruits and vegetables) and insoluble (wheat bran, vegetables, and whole grains) sources of fiber to the recommended 25–38 g/day is advisable [227]. Currently, there are no recommendations for fiber intake during the pandemic, but higher intakes may not be advisable due to the potential risk for gastrointestinal issues. 

### 5.3. General Dietary Advice during the COVID-19 Pandemic

While the Mediterranean diet, various food groups, and nutrients are just some of the examples of how nutrition plays an important role in modulating our immune health and preventing the development of NCDs, most international nutritional guidelines follow similar patterns, whereby they promote the high intake of fruits and vegetables and healthy quantities of the other food groups to provide all the essential nutrients for health. Hence, following any healthy diet at this time that provides an adequacy of RNIs will support immune function. However, there is concern that people who are deficient in some micronutrients such as vitamin C, vitamin D, or zinc may warrant supplementation or modify their dietary patterns to maintain a nutritional status and support healthy immune function. This is particularly important for our vulnerable populations, including the elderly and those with NCDs. Indeed, as many countries have imposed guidelines for people to “stay in place” and limit travel, etc., it is important for everyone to consider vitamin D levels when exposure to sunlight will be limited at this time [33,196]. Likewise, although not always possible depending on an individual’s personal circumstances, maintaining activity or moderate exercise at this time may be beneficial in supporting immune health [228]. Exercise in any form, even within the home, will help to maintain one’s health, as a lack of exercise could significantly affects a person’s cardiovascular risk in the future [106]. However, exercise may not be advisable for symptomatic COVID-19 patients, not only to reduce the transmission of the virus but, also, because some COVID-19 infections exhibit rare cardiac complications that could be aggravated by physical exertion [229]. 

Due to the “stay in place” guidelines, it is also important to consider boredom and stress at this time, as they are associated with increased food consumption, which can lead to people inadvertently seeking comfort during times of stress, like this current pandemic [126,230,231]. Sleep disturbances are also likely during isolation; thus, it is important to follow a normal sleep cycle if possible, along with promoting following normal eating patterns to promote the synthesis of melatonin and serotonin at dinner [126]. In particular, the WHO provides sound and practical advice to all on how to manage your health in terms of nutrition and wellbeing during self-quarantine [232].

It has been observed that many patients presenting to hospitals with COVID-19 infections have severe anorexia, inflammation, and markers of malnutrition. These additional vulnerabilities may increase risks for respiratory failure, necessitating noninvasive ventilation [83]. Furthermore, prolonged hospitalization may be accompanied by decreases in weight and muscle mass, which will further disadvantage at-risk patients [233,234]. Therefore, it is important to enhance nutrition to improve the response to therapy [235]. Two different Italian research consortia have proposed nutritional interventions: one to non-critically ill hospitalized COVID-19 patients [83] and the other for those who are hospitalized, severely ill, and may require enteral nutrition [235]. Both aim to increase the likelihood of beneficial clinical outcomes for hospitalized patients. Certainly, considering severe and nonsevere patient nutrition during hospitalization and interventions for those who recover from COVID-19 infections is of considerable importance and requires further research.

A final consideration to make is that there is a significant increase in pseudoscience and misinformation surrounding potential treatments and supplements to fight the COVID-19 infection. The authors would like to express again that there is currently no known effective treatment or mitigation strategy against COVID-19. Indeed, it has been recognized by others that there is considerable “emerging quackery” in relation to various foods and nutraceuticals with purported benefits and immune-enhancing supplements that may prevent or even cure the COVID-19 infection. This is symptomatic of levels of desperation and anxiety surrounding the crisis [236]. Consequently, it is important that pseudoscientific and false claims associated with nutritional products are avoided by manufacturers and reported to the relevant authorities to protect the health of those most vulnerable and susceptible within our communities. 

## 6. Conclusions

The COVID-19 pandemic is a significant threat to human life worldwide. As there is no known effective cure or treatment for COVID-19 yet, all potential therapeutics, mitigation interventions, and prevention strategies that may reduce the incidence or severity of infection are of vital importance. In this review, inflammation associated with pre-existing comorbidities was highlighted as a significant risk factor for COVID-19 patients. As a result, some of the anti-inflammatory therapies currently under investigation were discussed. Likewise, the potential role of a person’s nutritional status and nutrients and foods that may exert anti-inflammatory and immunomodulatory effects were explored, as summarized in Figure 3. Nutrients such as vitamin C, vitamin D, and zinc may hold some promise for the treatment of COVID-19. Likewise, nutrients with anti-inflammatory, antithrombotic, and antioxidant properties may prevent or attenuate the inflammatory and vascular manifestations associated with COVID-19. Indeed, following healthy dietary patterns and avoiding unhealthy dietary patterns, such as the Mediterranean and Western diets, respectively, may have beneficial effects against infection but require significantly more research. Our primary conclusion is that it is vitally important to maintain a healthy diet and lifestyle during the pandemic. This is especially important for the vulnerable in our society. In particular, those at high risks of infection should at least maintain their nutritional status by ensuring they reach their RDA of the potentially beneficial nutrients outlined in this review. Certainly, governments should actively encourage healthy eating and exercise in conjunction with the guidance provided on personal hygiene and physical distancing. That being said, various socioeconomic issues, lifestyle constraints, access to food, financial problems, or “stay in place” mitigation guidelines may reduce one’s ability to achieve this. Further research and review of the topics discussed in this manuscript are required as research evolves during the pandemic to discern the effective and ineffective strategies implemented globally. 

## Figures and Tables

**Figure 1 nutrients-12-01466-f001:**
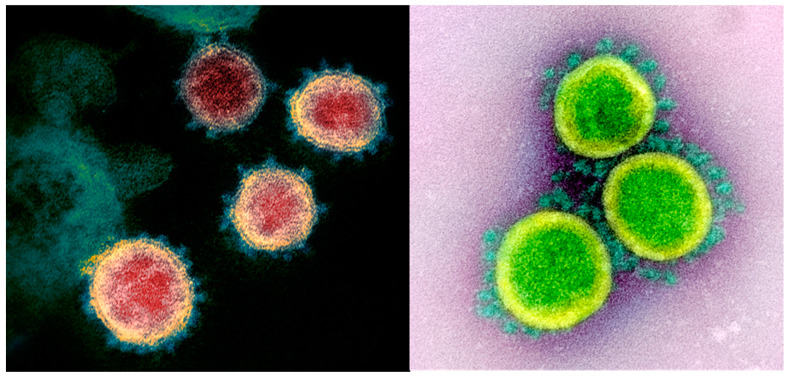
Images of SARS-CoV-2 obtained by transmission electron microscopy. The images show SARS-CoV-2 isolated from a patient in the United States. The virus particles shown are emerging from the surface of cells cultured in a laboratory. The spikes on the outer edge of the virus particles give coronaviruses their name. Both images were captured, colorized, and reproduced with permission courtesy of the National Institute of Allergy and Infectious Diseases’ Rocky Mountain Laboratories in Hamilton, MT, USA [23].

**Figure 2 nutrients-12-01466-f002:**
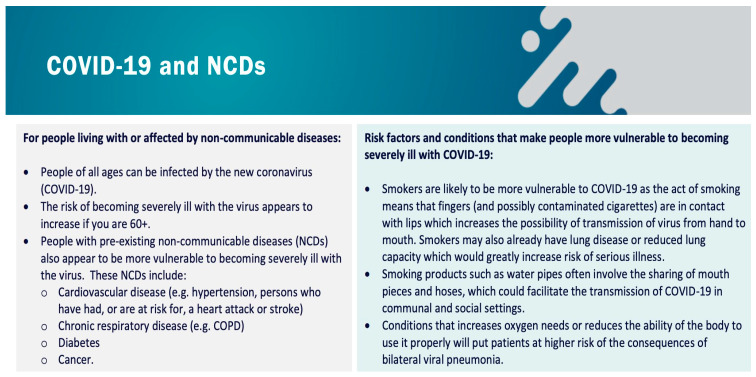
World Health Organization (WHO) guidelines and information intended for people with noncommunicable diseases [25]. Abbreviations: COPD = Chronic obstructive pulmonary disease.

**Figure 3 nutrients-12-01466-f003:**
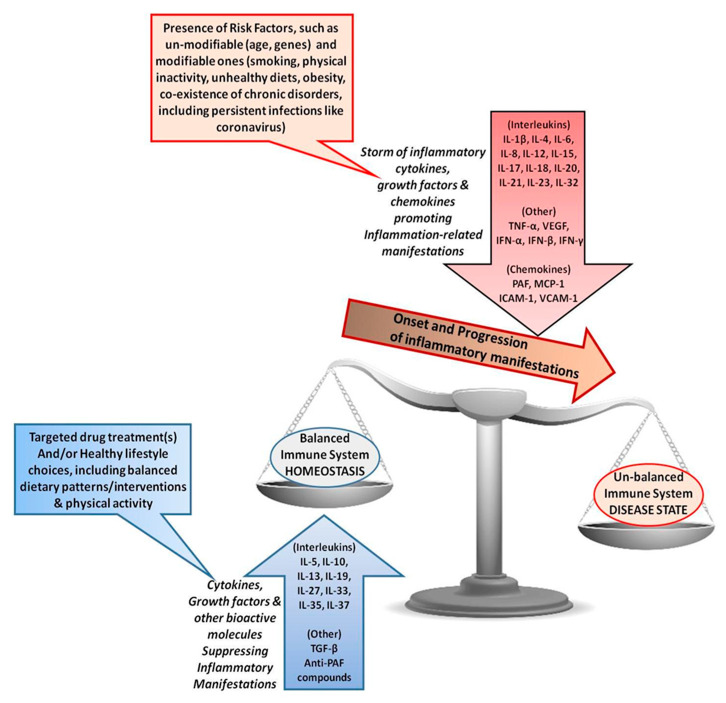
A representation of the inflammatory molecules involved in infection and how certain dietary components may interact with them. Included are the cytokines, growth factors, and other chemokines that promote the onset and progression of inflammatory manifestations, dysregulation of the immune system, and a diseased state due to the effects of several unmodifiable and modifiable risk factors. Targeted drug treatments and/or healthy lifestyle choices such as balanced diets and physical activity can counterbalance such inflammatory manifestations towards homeostasis.

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
