# Peer review of "COVID-19: The Inflammation Link and the Role of Nutrition in Potential Mitigation"

_nutrients, 2020, doi:10.3390/nu12051466_

Round 1

Reviewer 1 Report

The review is well written but from a nutritional point of view, it doesn’t give any information about  the mechanism by diet and/or supplement by specific nutrients or pool of them that could modulate inflammation and consequent (and less mentioned) ROS production.

342-362 the authors mention bioactive compounds but they don’t elucidate which one and how they could act on fighting COVID or help to manage the disease

364-377 western diet is a possible problem but the authors don’t consider the first problem i.e.: hyperglycemia and its consequences such as AGEs, RAGEs and so on, which are strongly inflammatory and detrimental for the immune system

379-387 correct suggestion, but the most important class of substances i.e. polyphenols is not considered

389-419 the mechanism on modulating prostaglandin is not elucidated and the possible effect on blood aggregation is not considered

421-443 since inflammation is the source of ROS it is evident that vitamin C, not as RDA, could help. In effect, it’s reported a dosage ranging from 1 to 4.5 grams per day in COVID patient

Conclusion: the conclusion is too general. It is not easy to give advice but at least a range of possible doses of the nutrients and supplements mentioned is expected. The authors provide only a small tip for the fibers, but this is the suggested normal intake, therefore not "adapted" to COVID patients

Author Response

Comments and Suggestions for Authors

Rev 1: The review is well written but from a nutritional point of view, it doesn’t give any information about  the mechanism by diet and/or supplement by specific nutrients or pool of them that could modulate inflammation and consequent (and less mentioned) ROS production.

Au: We would like to thank the reviewer for their constructive comments on this review. We appreciate their insights on the manuscript.

It should be noted that the anti-inflammatory, anti-ROS, and antithrombotic mechanisms of several healthy nutrients/diets in general have previously thoroughly been described in several reviews that are widely referenced in the present paper. We have purposely not gone too deeply into the mechanisms for each nutrient as they are wide and vast and the review would become extremely large. However, where a mechanism may be specific to COVID-19, such as the additions to the fish oil section, we have discussed the potential mechanisms at play in further detail. It should also be mentioned that as SARS-CoV-2 is a novel virus, there is a lack of significant evidence to provide direct mechanistic explanations for any of the nutrients as the nutrients themselves are still under investigation. At this point in time it would be naïve of us to proclaim any mechanism for any nutrient.

We have added additional mechanisms related to the detrimental effects of COVID-induced inflammatory manifestations, which can putatively beneficially affected by specific nutrients and diets. Thus, relative texts and references have also been added throughout the manuscript to support the mechanisms proposed for these nutrients/diets against such viral-induced manifestations of inflammation, thrombosis and ROS.

Rev 1: 342-362 the authors mention bioactive compounds but they don’t elucidate which one and how they could act on fighting COVID or help to manage the disease

Au: We have added a paragraph mentioning some bioactive compounds and proposed mechanisms of their protective and preventive effects against inflammation, thrombosis and ROS that can putatively attenuate such virus-induced manifestations. However, as before, SARS-CoV-2 is a novel virus, as a result it would be incorrect at this time to narrow down to any specific nutrient currently as there simply is not enough evidence to support such a claim right now. Evidence for their effects may take several months or years. As demonstrated in the vitamin C section, there are currently trials underway. However, their termination is not due until 2021. Thank you for this suggestion.

Rev 1: 364-377 western diet is a possible problem but the authors don’t consider the first problem i.e.: hyperglycemia and its consequences such as AGEs, RAGEs and so on, which are strongly inflammatory and detrimental for the immune system

Au: Thank you for this suggestion, we have now included the role of hyperglycaemia and AGEs in inflammation, but also as a risk factor in COVID-19, supported with relevant literature.

Rev 1: 379-387 correct suggestion, but the most important class of substances i.e. polyphenols is not considered

Au: We have added polyphenols as a potential bioactive compound in this section; thank you kindly for your suggestion.

Rev 1: 389-419 the mechanism on modulating prostaglandin is not elucidated and the possible effect on blood aggregation is not considered

Au: We have now added specific text and related references in this section concerning the implication of prostaglandins in COVID and subsequent effects of bioactive fish lipids. In addition the potential beneficial anti-inflammatory and antithrombotic properties of fish bioactive lipids and peptides against platelet aggregation induced by several pathways has also been further discussed. Thank you.

Rev 1: 421-443 since inflammation is the source of ROS it is evident that vitamin C, not as RDA, could help. In effect, it’s reported a dosage ranging from 1 to 4.5 grams per day in COVID patient

Au: Thank you for this suggestion, we could not find any documents supporting a dose of 4.5 g, however, where there are suggested increased doses of vitamins and minerals we have now included those at the end of each section. Thank you for your constructive comment.

Rev 1: Conclusion: the conclusion is too general. It is not easy to give advice but at least a range of possible doses of the nutrients and supplements mentioned is expected. The authors provide only a small tip for the fibers, but this is the suggested normal intake, therefore not "adapted" to COVID patients

Au: Our conclusion cannot recommend nutrient intakes in excess of current RDAs due to the absence of a sufficient evidence base in support of higher doses. The requirement for intakes in excess of RDAs has not been established either in COVID-19 patients, or for members of the public who may wish to attenuate risk by optimising nutritional status. Until reference intakes are established for these unprecedented scenarios, public health recommendations remain at the attainment of RDAs specific to the individuals’ age, gender, pre-existing medical conditions and jurisdiction. These are outlined with the text. Overall, we thank you very much for your insightful advice. Your review has certainly improved the manuscript.

Reviewer 2 Report

The title of this article is interesting and it arouses interest, but unfortunately it is not fully reflected in the presented thesis. The first part gives current data on COVID-19 infection, followed by a description of the mechanisms responsible for the dynamic, strong response of the immune system and its consequences. The authors wanted to demonstrate changes at the cellular level, the contribution of individual signaling pathways, and systemic consequences of all the above.

In chapter 3, the authors generally described attempts at pharmacological modification of the patient's immune response to COVID-19. Finally, since chapter 4, they tried to find a connection between response of the immune system to COVID-19 infection and associate it with nutrition. At the same time, it is important to remember, that this particular information can be universally included in almost any viral infection. These data are widely known, proper nutrition is an important factor in the functioning of the body, and the Mediterranean diet is rich in „antioxidants”, so it is peculiarly promoted around the world. Likewise, it should be taken into consideration, whether in the current situation, not only eating habits of the Mediterranean basin (Italy, Spain) are a good example of behavior to follow, regardless of the type of diet, where despite the exemplary diet, they are still struggling with a developed of epidemiological situation and mortality is significantly higher compared to the northern regions of Europe. In the contrary, Western countries with nutrition based on processed, high-calorie products, which are rich in fats and carbohydrates, seem to better „endure” the dynamics and progress of COVID infection. So, it is very difficult to draw any conclusions for today.

Obviously, promotion of the widely regarded „healthy nutrition” is desirable, because it is undoubtedly associated with the improvement of public health in the broad sense (reduction of incidence of diabetes, cardiovascular disease and many others). Comparison of specific data from malnourished patients (without comorbidities) with patients who have a proper nutritional condition and both are suffering from COVID-19 infection could in some very indicative way develop the problem contained in the topic of the article.

In my opinion, the current version of the article raises the problem of nutrition in various diseases, including viral infection.

Author Response

Comments and Suggestions for Authors

Rev 2: The title of this article is interesting and it arouses interest, but unfortunately it is not fully reflected in the presented thesis. The first part gives current data on COVID-19 infection, followed by a description of the mechanisms responsible for the dynamic, strong response of the immune system and its consequences. The authors wanted to demonstrate changes at the cellular level, the contribution of individual signaling pathways, and systemic consequences of all the above. In chapter 3, the authors generally described attempts at pharmacological modification of the patient's immune response to COVID-19. Finally, since chapter 4, they tried to find a connection between response of the immune system to COVID-19 infection and associate it with nutrition. At the same time, it is important to remember, that this particular information can be universally included in almost any viral infection. These data are widely known, proper nutrition is an important factor in the functioning of the body, and the Mediterranean diet is rich in „antioxidants”, so it is peculiarly promoted around the world.

Au: First, thank you kindly for your review of our manuscript. We are happy to hear you found the title interesting. The reviewer has outlined the content of our manuscript but has expressed that the content does on reflect the title. The title of our manuscript “COVID-19 and Chronic Diseases: The Inflammation Link and the Role of Nutrition in Potential Mitigation” We have endeavoured to provide a section for each of the key words in the title and within each section we make reference to the underlying link between these factors affecting COVID-19, which is inflammation. We have modified the title to drop “and chronic diseases” in order to highlight the main points of the paper, which are inflammation, anti-inflammatory strategies, and nutrition.

To your last point, “that this particular information can be universally included in almost any viral infection” yes this is correct simply because we do not know the specifics of the pathogenicity of SARS-CoV-2, therefore we cannot declare that one specific food or nutrient is effective. It will take a lot more research to determine what nutrients may be effective. This is similar to the fact that there are greater than 102 vaccines under development, if vaccine scientists knew everything they would only have tried to develop 1 or 2 vaccines. Nutrition research is the same, the best and most scientifically sound advice at this moment is to obtain nutrients associated with improved immune function.  You could also look at the vitamin D story, one study in Switzerland says low vitamin D levels are associated with COVID-19 infection, whereas the UK biobank study finds the opposite. This review provides an overall scope of the nutrition research in relation to COVID-19.

Rev 2:

Likewise, it should be taken into consideration, whether in the current situation, not only eating habits of the Mediterranean basin (Italy, Spain) are a good example of behavior to follow, regardless of the type of diet, where despite the exemplary diet, they are still struggling with a developed of epidemiological situation and mortality is significantly higher compared to the northern regions of Europe. In the contrary, Western countries with nutrition based on processed, high-calorie products, which are rich in fats and carbohydrates, seem to better „endure” the dynamics and progress of COVID infection. So, it is very difficult to draw any conclusions for today.

Au: To your point implying countries following a Western diet are enduring or doing better against COVID-19, this is not the case and there is no evidence at this point to suggest that. There are many factors that affect the infection rates in different countries. Also, countries are currently at different stages of disease progression as it is an evolving situation dependent on the success of public health mitigation strategies. For instance, Italy was the first country in Europe to have a significant number of cases it now currently has a case fatality rate of 13.9%, for comparison France (14.9%), Ireland (6.3%), UK (14.7%), Greece (5.6%), Spain (11.8%), Turkey (2.7%), Portugal (4.1%), Croatia (4.0%). As you can see some Mediterranean countries such as Greece have lower case fatalities than others. It is also important to note that it can take a long time for death to occur from COVID-19, so these case fatality rates are not entirely true of the situation and can’t be used as evidence until the pandemic is over. The United States currently has 1.3 million confirmed cases and 78,000 dead, it is projected that another 22,000 may die, therefore the case fatality rate will significantly rise. Population dynamics and demographics, age, comorbidities, nutritional status, socioeconomic status, mitigation strategies, the progression of COVID-19 all contribute to how the disease spreads. Likewise, mortality levels depend on the level of testing, contact tracing and the countries health systems.

We are not suggesting that the countries of the Mediterranean are doing better. We have presented that the Med diet as a dietary pattern with significant evidence supporting its anti-inflammatory potential that may convey beneficial effects. As stated from lines 396-398 “Of course, it is the ideals of the Mediterranean diet that are important to follow, as a traditional Mediterranean diet is practically non-existent these days and countries of the Mediterranean follow a diet more closely related to the Western diet than that of their ancestors [128].” Here we highlight that the Mediterranean diet is currently not consumed widely as France, Italy, Spain, etc. all resemble lifestyles characterised by a Western diet now due to modernisation and social factors such as urbanisation.

Rev 2: Obviously, promotion of the widely regarded „healthy nutrition” is desirable, because it is undoubtedly associated with the improvement of public health in the broad sense (reduction of incidence of diabetes, cardiovascular disease and many others). Comparison of specific data from malnourished patients (without comorbidities) with patients who have a proper nutritional condition and both are suffering from COVID-19 infection could in some very indicative way develop the problem contained in the topic of the article. In my opinion, the current version of the article raises the problem of nutrition in various diseases, including viral infection.

Au:

We thank the reviewer for their constructive feedback. However, it is important to note that the pandemic is still evolving and so research regarding the role of nutrition in COVID-19 is almost non-existent. Therefore, the aim of this review is to provide evidence-based guidance using our knowledge of previous and viral infections such as MERS, SARS, influenza, and other related viruses in order to determine what nutritional strategies that might reduce the risk/severity of infections. However, as you have pointed out nutrition can play a role in various diseases. That is the point we are trying to make. A healthy diet coincidently reduces the risk of developing NCDs and may improve disease states, which may be beneficial against COVID-19 due to the greater risk of infection when a patient has a pre-existing NCD. As a consequence of maintaining adequate nutritional status it also promotes a healthy immune system, which indeed could affect multiple viral infections. This is important to document considering the strong link between ageing individuals and those with comorbidities to COVID-19 infection. However, to focus this review, we have narrowed down our approach to some key nutrient groups that are particularly associated with the immune response and a reduction in the severity of other viral infections such as influenza. Indeed, since your review of the manuscript we have also added evidence-based dosing guidance where appropriate and available to enhance the relevance of the review in accordance with the other reviewer’s suggestions.

Rev 2: In response to the following reviewer comment: “Comparison of specific data from malnourished patients (without comorbidities) with patients who have a proper nutritional condition and both are suffering from COVID-19 infection could in some very indicative way develop the problem contained in the topic of the article”

Au: We agree with this completely, but unfortunately there is no specific data or studies ongoing in relation to the nutritional status of malnourished infected patients versus those infected with adequate nutritional status. However, guidelines cited in this review are proposing the inclusion of assessing the nutritional status of a COVID patients on admission to hospital (section 5.4), which might pave the way for such studies.  

Round 2

Reviewer 2 Report

The improved form of this manuscript is suitable for publication.